# Combination Treatment of a Phytochemical and a Histone Demethylase Inhibitor—A Novel Approach towards Targeting TGFβ-Induced EMT, Invasion, and Migration in Prostate Cancer

**DOI:** 10.3390/ijms24031860

**Published:** 2023-01-17

**Authors:** Nidhi Dalpatraj, Ankit Naik, Noopur Thakur

**Affiliations:** Department of Biological and Life Sciences, School of Arts and Sciences, Ahmedabad University, Ahmedabad 380009, India

**Keywords:** GSK-J4, epithelial-to-mesenchymal transition, combinatorial treatment, drug synergism, epigenetics

## Abstract

Minimizing side effects, overcoming cancer drug resistance, and preventing metastasis of cancer cells are of growing interest in current cancer therapeutics. Phytochemicals are being researched in depth as they are protective to normal cells and have fewer side effects. Hesperetin is a citrus bioflavonoid known to inhibit TGFβ-induced epithelial-to-mesenchymal transition (EMT), migration, and invasion of prostate cancer cells. Targeting epigenetic modifications that cause cancer is another class of upcoming therapeutics, as these changes are reversible. Global H3K27me3 levels have been found to be reduced in invasive prostate adenocarcinomas. Combining a demethylase inhibitor and a known anti-cancer phytochemical is a unique approach to targeting cancer to attain the aforementioned objectives. In the current study, we used an H3K27 demethylase (JMJD3/KDM6B) inhibitor to study its effects on TGFβ-induced EMT in prostate cancer cells. We then gave a combined hesperetin and GSK-J4 treatment to the PC-3 and LNCaP cells. There was a dose-dependent increase in cytotoxicity and inhibition of TGFβ-induced migration and invasion of prostate cancer cells after GSK-J4 treatment. GSK-J4 not only induced trimethylation of H3K27 but also induced the trimethylation of H3K4. Surprisingly, there was a reduction in the H3K9me3 levels. GSK-J4 alone and a combination of hesperetin and GSK-J4 treatment effectively inhibit the important hallmarks of cancer, such as cell proliferation, migration, and invasion, by altering the epigenetic landscape of cancer cells.

## 1. Introduction

Prostate cancer occurs in the prostate gland located just beneath the urinary bladder. Factors, such as age, ethnicity, and family history, are associated with prostate cancer occurrence. In India, prostate cancer is among the top ten leading types of cancer. In a report by the National Cancer Registry Programme, India, for the year 2020, prostate cancer is among the five most common cancers in men. The projected incidence rate was 41,532 cases with a cumulative risk of 1 in 125 [1]. The incidence rate of prostate cancer ranges from 5.0 to 9.1 per 100,000/year, with 85% of the cases being detected at the late stages (III and IV) [2]. The number of cases is increasing year by year. It is predicted that 268,490 cases will be newly diagnosed in 2022 with 3.5 million men having a past diagnosis [3]. Existing treatment strategies for prostate cancer include surgery, radiation, hormone therapy, chemotherapy, and vaccination. However, most of them are either associated with serious side effects or are ineffective after the cancer has metastasized. Therefore, it is important to develop treatment strategies with minimum side effects that can target and prevent cancer metastasis.

Cancer cells metastasize by undergoing epithelial-to-mesenchymal transition (EMT). Usually, this process is required during embryogenesis and wound healing. The cells change their morphology from tightly adhered epithelial to loosely attached mesenchymal phenotypes. The epithelial cells are held tightly by proteins, such as E-cadherin and occludins. When these proteins are downregulated, the cells lose cell-to-cell and cell-to-matrix contact. They attain the mesenchymal phenotype by upregulating the expression of proteins, such as N-cadherin and vimentin [4]. The mesenchymal cells are spindle-shaped and are free to migrate and metastasize. Type 3 EMT is responsible for characteristics, such as invasion and migration of cancer cells, leading to the metastasis of cancer [5]. Most cases have a poor prognosis because the cancer cells have metastasized. Targeting this process is, therefore, important.

TGFβ is a cytokine that is well known for its role in inducing EMT in cancer cells [6]. It regulates EMT via both its canonical and non-canonical pathways [6]. The canonical TGFβ pathway involves signalling mediated via the Smad proteins. There are eight smad proteins, and they can be classified as receptor-regulated Smads (R-Smads; Smads 1, 2, 3, 5, and 8); the common mediator of Smad (Co-Smad), Smad4; and inhibitory Smads (I-Smads; Smad6 and Smad7) [7]. The non-canonical pathways involve, among others, signalling mediated by mitogen-activated protein kinase (MAPK), Hippo, phosphoinositide 3-kinase (PI3K)/AKT, and AMP-activated protein kinase (AMPK) signaling [8]. The downstream signalling of these pathways results in the transition from epithelial to mesenchymal phenotypes by regulating the expression of transcription factors, such as Snail, ZEB, and bHLH families. TGFβ induces the expression of these transcription factors, which, in turn, downregulate the expression of epithelial genes and upregulate mesenchymal gene expression [9]. In a novel study, it has been reported that TGFβ via Smad7 increases the levels of c-Jun and HDAC6, thereby enhancing prostate cancer cell invasion and migration [10]. We, therefore, used TGFβ to induce EMT in prostate cancer cells. To look at the effect of drug treatments on TGFβ-induced EMT, we looked at the expression of p-Smad (for canonical signaling), p-c-Jun (for non-canonical signaling), N-cadherin, E-cadherin, and vimentin (for markers of EMT).

Although all the cells in an individual contain the same genetic material, they differ in their functions because of differential gene expression. This is mainly controlled by epigenetic modifications, which include DNA methylation, histone methylation, histone acetylation, and non-coding RNAs [11]. Many epigenetic changes are involved in the process of EMT. In breast tumor cells, it was found that CDH1 promoter hypermethylation caused the reduced expression of E-cadherin rather than its mutational inactivation [12]. Enzymes, such as SUV39H1 and SETDB1 (lysine methyltransferases), have been suggested as potential therapeutic targets as they have been shown to enhance prostate cancer cell migration and invasion [13]. Another study identified 32 enzymes that belonged to the family of JmjC-domain-containing histone demethylases as being critical for prostate cancer proliferation and survival [14]. Epigenetic changes, unlike changes in DNA sequences, are reversible. Targeting and inhibiting these enzymes have been shown to have promising anti-cancer effects in cancer cells.

We checked the levels of different histone modifications in the PC3 cells and found that the trimethylation of H3K27 was highly downregulated. Therefore, we used GSK-J4, a histone H3K27 demethylase (JMJD3/KDM6B) inhibitor, to increase the H3K27me3 levels to check if it leads to prostate cancer progression inhibition. GSK-J4 has been shown to have therapeutic potential for acute myeloid leukemia (AML), glioma, neuroblastoma, and breast cancer [15,16,17,18]. The trimethylation of H3K27 (H3K27me3) leads to the inactivation of gene expression. It has been reported that in invasive adenocarcinomas and prostatic intraepithelial neoplasia, the global H3K27me3 levels were reduced [19]. Additionally, decreased H3K27me3 levels correlated with increased Gleason score [19]. In ovarian cancer cells, it was seen that the removal of the H3K27me3 marks catalyzed by EZH2 was required for the transition to a mesenchymal state [20]. KDM6B was also found to promote tumorigenesis in prostate cancer cells by inducing transcription of cyclin D1 and knocking it out was shown to have anti-cancer effects [21]. We, therefore, used GSK-J4—a histone demethylase inhibitor—to target TGFβ-induced EMT in prostate cancer cells.

One approach to minimizing the side effects of anti-cancer drugs is the use of phytochemicals. Several natural compounds, such as curcumin, resveratrol, and gallocatechins, have been found to have anti-cancer properties [22]. Since phytochemicals have multiple targets and lower systemic toxicity, they are more advantageous for cancer prevention and treatment, as compared to drugs with single molecular targets [23]. In our previous study, we showed that hesperetin, a citrus bioflavonoid, could induce cell cycle arrest and inhibit TGFβ-induced EMT, invasion, and migration of prostate cancer cells [24].

Since GSK-J4 also showed anti-cancer properties, we combined hesperetin and GSK-J4 and checked for synergism between both compounds. Two compounds are said to be synergistic if their combined effect is more significant than their individual effect. Combinatorial treatments are often preferred over monotherapy as it slows down the chances of developing drug resistance. It also reduces individual drug doses, which will, in turn, reduce drug toxicity to normal cells, thereby minimizing side effects.

## 2. Results

### 2.1. GSK-J4 Treatment Showed a Dose-Dependent Increase in Cell Cytotoxicity in Prostate Cancer Cells

MTT assay was carried out to check the effect of GSK-J4 on the proliferation of prostate cancer cells. The PC-3 cells were treated with different concentrations of GSK-J4 ranging from 1 µM to 100 µM. There was a dose-dependent decrease in cell proliferation both after 24 and 48 h of treatment, ranging from a 50% decrease in cell viability at 20 µM and up to 10% viability at 100 µM in the PC-3 cells (Figure 1a). Trypan blue assay was carried out to confirm the cytotoxicity of GSK-J4, and a dose-dependent increase in cell cytotoxicity was observed (Figure 1b).

### 2.2. GSK-J4 Inhibits TGFβ-Induced EMT in Prostate Cancer Cells

Western blot analysis demonstrated no significant difference in the expression of EMT markers after a treatment of 24 h (Figure 1c). We, therefore, extended the GSK-J4 treatment to 48 h and observed a decrease in the expression of N-cadherin and vimentin (Figure 1(d)). We observed an increase in the levels of E-cadherin transcript after GSK-J4 treatment for 48 h, both in the presence and absence of TGFβ (Figure 1d). We witnessed a decrease in the p-Smad3 levels, whereas there was no change in p-c-Jun levels after GSK-J4 treatment in the presence of TGFβ for 48 h, suggesting that GSK-J4 inhibits the TGFβ canonical signaling in PC-3 cells (Figure 1d). In all, the effect of GSK-J4 was more pronounced in TGFβ-induced cells, as compared to GSK-J4 treatment without TGFβ induction. The quantification of all the Western blot images can be found in Appendix A File S2.

### 2.3. GSK-J4 Significantly Inhibits TGFβ-Induced Migration and Invasion in Prostate Cancer Cells

Wound-healing assay was carried out to determine the effect of GSK-J4 on the migratory ability of PC-3 cells. GSK-J4 significantly inhibited cell migration in the presence of TGFβ, as compared to TGFβ treatment alone (Figure 1e). Only about 35–40% of the wound closed after GSK-J4 treatment compared to TGFβ treatment alone (Figure 1e). GSK-J4 also significantly reduced the invasiveness of TGFβ-stimulated cells, as studied using the Boyden chamber assay. Only about 100 cells had invaded across the membrane, as compared to almost 600 cells in the TGFβ-treated cells (Figure 1f). Despite the high tumorigenicity and high metastatic potential of PC-3 cells, GSK-J4 effectively inhibited the migratory and invasive potential of these cells.

### 2.4. Effect of GSK-J4 on Histone Modifications

Since GSK-J4 is an H3K27 demethylase inhibitor, an accumulation of H3K27me3 levels is expected after GSK-J4 treatment. Moreover, Kruidenier et al. showed that GSK-J4 not only inhibits KDM6A/6B but also inhibits KDM5B and KDM4C in vitro and in tissue culture assays [25]. We, therefore, looked at the levels of H3K27me3, as well as H3K4me3 and H3K9me3, after treatment with GSK-J4. When we looked at the trimethylation levels on different lysine residues of histone H3, we found an accumulation of both H3K4me3 and H3K27me3 levels after treatment with GSK-J4 in the presence of TGFβ, but surprisingly there was a decrease in the H3K9me3 levels (Figure 1g). Therefore, it is evident that GSK-J4 does not work as a specific inhibitor of H3K27 demethylase. We also verified the results by looking at the transcripts levels of the respective enzymes as well. The transcript levels of MLL1 (H3K4 methylase) and EZH2 (H3K27 methylases) had increased, whereas SUV39H2 (H3K9 methylase) transcripts decreased after treatment with GSK-J4 in the presence of TGFβ (Figure 1h).

### 2.5. Effect of GSK-J4 on the Proliferation, EMT, Histone Modifications, Migration, and Invasion of LNCaP—An Androgen-Sensitive Cells

After checking the effect of GSK-J4 on the hormone-resistant and aggressive cell line (PC-3), we studied the effect of GSK-J4 on an androgen-sensitive cell line that is less tumorigenic (LNCaP). It was found that the IC50 for LNCaP cells at 24 h was at a concentration of about 30 µM, and at 48 h, the IC50 was at around 20 µM. At 50 µM, the cell viability was reduced to 15% (Figure 2a). There was, therefore, a dose-dependent increase in cytotoxicity in LNCaP cells after GSK-J4 treatment.

We checked the protein expression of epithelial and mesenchymal markers to study the effect of GSK-J4 on TGFβ-induced EMT. After 48 h of GSK-J4 treatment, an increased expression of the epithelial marker E-cadherin at the transcript level was observed (Figure 2b). There was also a decrease in the expression of the mesenchymal markers N-cadherin and vimentin compared to TGFβ treatment (Figure 2b). In the LNCaP cells, there was no difference in the p-Smad3 levels after GSK-J4 treatment in the presence of TGFβ; however, p-c-Jun levels decreased, suggesting that the TGFβ signaling is inhibited via the non-canonical pathway in LNCaP cells (Figure 2b). As observed in the PC-3 cells, the effect of GSK-J4 was more pronounced in TGFβ-induced cells, as compared to GSK-J4 treatment without TGFβ induction.

A wound healing assay and Boyden chamber assay were carried out to look at the effect of GSK-J4 on the migratory and invasive potential of LNCaP cells. In LNCaP cells, only about 2% of the wound had closed in the presence of GSK-J4 treatment (Figure 2c). There was almost a 50% reduction in the migration of cells treated with GSK-J4 in the presence of TGFβ, as compared to TGFβ treatment alone (Figure 2c). The invasion assay revealed that none of the cells had invaded across the membrane after GSK-J4 treatment, both in the presence and absence of TGFβ (Figure 2d). Since the LNCaP cells have low tumorigenicity, it shows increased sensitivity to GSK-J4 treatment.

The change in all three histone modifications (H3K4me3, H3K9me3, and H3K27me3) targeted by GSK-J4 in the LNCaP cells was similar to those observed in the PC-3 cells. There was an increase in the H3K4me3 and H3K27me3 levels and a decrease in H3K9me3 levels after GSK-J4 treatment (Figure 2e). There was an increase in the transcript levels of both MLL1 (H3K4 methylase) and EZH2 (H3K27 methylase) enzymes. The levels of SUV39H2 (H3K9 methylase) transcripts were downregulated when GSK-J4 treatment was given in the presence of TGFβ, as compared to its levels in TGFβ-induced cells (Figure 2f).

### 2.6. Effect of Hesperetin, GSK-J4, and Their Combination on Prostate Cancer Cells

MTT assay was carried out to check the effect of hesperetin and GSK-J4 alone and in combination on the proliferation of PC-3 and LNCaP cells. The obtained IC50 was around 100 µM of hesperetin and 10 µM of GSK-J4 when given in combination for PC-3 cells (Figure 3a), whereas for LNCaP cells, the dose was further reduced to 40 µM of hesperetin and 4 µM of GSK-J4 (Figure 4a). Combining both compounds reduced their individual doses significantly.

### 2.7. Combination Index (CI) of Hesperetin and GSK-J4 to Infer Synergism, Additivity, or Antagonism Using Compusyn Software 1.0

Both hesperetin and GSK-J4 were found to be cytotoxic to prostate cancer cells. To understand the type of interaction between the two compounds when used in combination, the Chou–Talalay method was used for synergy quantification. Compusyn software for dose–effect analysis is based on the “mass-action law.” The software computes the “combination index (CI)” value, which quantitatively measures the synergism (CI < 1), additive effect (CI = 1), and antagonism (CI > 1).

The IC50 for the combination of hesperetin and GSK-J4 in PC-3 cells was found to be 121.696 µM, where the individual concentration of hesperetin and GSK-J4 was 110.633 µM and 11.0633 µM, respectively (Figure 3a). At concentrations ranging from 10 µM to 150 µM, there was a synergistic relationship between hesperetin and GSK-J4 (Figure 3b). In the concentration range of 200 µM to 250 µM, there was a nearly additive effect, and at concentrations above 250 µM, the combination was found to be antagonistic (Figure 3b). Another objective of using drug combinations was to decrease the dosage of individual drugs, for which the concept of the dose-reduction index (DRI) was used. DRI > 1 indicates favorable dose reduction, DRI = 1 indicates no dose reduction, and DRI < 1 indicates non-favorable dose reduction. In our study, the DRI was found to be favorable at concentrations where up to 75% of cells were affected (Figure 3b). The DRI was not favorable at higher concentrations, where 80 to 97% of the cells were affected. (Figure 3b).

In the LNCaP cells, the IC50 for the drug combination was at 40.1 µM, where the individual concentration of hesperetin and GSK-J4 was 36.4550 µM and 3.64550 µM, respectively (Figure 4a). Almost all the drug combinations showed a synergistic effect, and the highest combination of drugs gave a nearly additive effect on the LNCaP cells (Figure 4b). The DRI values were also favorable in all the combinations. This again shows that the combination treatment was more effective in the LNCaP cells (Figure 4b). The detailed Compusyn report for both cell lines are available as Appendix A.

### 2.8. A Combination of Hesperetin and GSK-J4 Inhibits TGFβ-Induced EMT by Inhibiting Both Canonical and Non-Canonical Signaling Pathways in Both Cell Lines

The EMT markers, such as E-cadherin, N-cadherin, and vimentin, were studied using Western blot analysis. In both the cell lines, the expression of E-cadherin at the transcript level increased when treated with the drug combination. In contrast, the expression of N-cadherin and vimentin decreased at the protein level when the drug combination was given in the presence of TGFβ (Figure 3c and Figure 4c). Moreover, the levels of both p-Smad3 and p-c-Jun decreased when the drug combination was given, suggesting that hesperetin and GSK-J4 together inhibit both the canonical and non-canonical TGFβ signaling pathways in the PC-3 and LNCaP cells (Figure 3c and Figure 4c).

### 2.9. Effect of Combinatorial Treatment of Hesperetin and GSK-J4 on Migration and Invasion of Prostate Cancer Cells

A wound healing assay was carried out to check the effect of the compound alone at a lower concentration and in combination on the migratory potential of the cells. The experiment showed that 100 µM of hesperetin alone and 10 µM of GSK-J4 independently did not significantly inhibit TGFβ-induced migration, but a combination of those significantly inhibited the migratory potential of PC-3 cells and a combination of 40 µM of hesperetin and 4 µM of GSK-J4 in that of LNCaP cells (Figure 3d and Figure 4d). Similarly, transwell chambers with Matrigel were used to check the effect on the invasive capability of cells. It was concluded that in combination, hesperetin and GSK-J4 reduce the invasiveness of PC-3 and LNCaP cells at a much lower dosage, as compared to attaining similar effects when used individually (Figure 3e and Figure 4e).

### 2.10. Effect of Combinatorial Treatment of Hesperetin and GSK-J4 on Histone Modifications

We checked the effect of combinatorial treatment on the levels of H3K4, H3K9, and H3K27 trimethylation in both cell lines using Western blot analysis. It was found that in the PC-3 cells, there was not a significant difference in the H3K4me3 levels. However, H3K9me3 and H3K27me3 increased with the treatment of hesperetin and GSK-J4 (Figure 3f).

In the LNCaP cells, again, there was not a significant difference in the H3K4me3 levels across different treatments. In contrast to the PC-3 cells, the H3K9me3 levels decreased significantly in the LNCaP cells, after the combinatorial treatment. The H3K27me3 levels increased when hesperetin and GSK-J4 treatment was given in the presence of TGFβ (Figure 4f).

## 3. Discussion

Prostate cancer has a high recurrence rate as the cancer cells are difficult to be targeted after they have metastasized to the bone or lymph node. In an autopsy performed on 631 patients with stage IV prostate cancer, 90% had bone metastases, and lymph nodes were involved in 66% of the patients [26]. It thereby becomes important to focus on treatment strategies that target and prevent prostate cancer metastasis. Since the mutations in the DNA sequence that lead to cancer progression are irreversible, we can focus on the epigenetic changes involved in the malignancy as they are reversible.

The inhibition of JMJD3 (H3K27 demethylase) led to decreased cell viability by inducing apoptosis and reduced cell migration in a glioma cell line [16]. In a study on the NMuMG cell line, they found that TGFβ-induced loss of E-cadherin expression coincided with reduced H3K9Ac and H3K4me3 and an increase in H3K27me3. When they withdrew TGFβ, it restored E-cadherin levels along with H3K9Ac and H3K4me3 modifications [27]. Another study reported that the inhibition of H3K27me3 demethylases suppressed migration and invasion in breast cancer cell lines by the inactivation of CEMIP (cell-migration-inducing hyaluronidase) [18]. Increased H3K9me3 has been reported in invasive colorectal cancer tissues and showed a positive correlation with lymph node metastasis in CRC patients [28].

We observed increased levels of H3K4me3 and H3K27me3 and a decrease in H3K9me3 when GSK-J4 treatment was given in the presence of TGFβ. Since H3K4me3 is an active mark, we will look more deeply at the genes associated with this mark. We expect the tumor suppressor genes to become associated and activated to suppress the TGFβ induced EMT. The increase in H3K27me3 after GSK-J4 treatment was expected as GSK-J4 is an inhibitor of the H3K27 demethylase enzyme. Moreover, increased H3K27me3 levels have been reported in glioma and breast cancer cell lines to be involved in reducing the migratory capability of cancer cells, thereby supporting our data.

There was also a drastic decrease in H3K9me3 levels after GSK-J4 treatment in the presence of TGFβ. H3K9me3 marks are associated with the repair of double-stranded DNA breaks [29] and also with maintaining genome stability and the heterochromatin state of DNA [30]. Further studies are required to understand the role of decreased H3K9me3 levels in GSKJ4-treated cells and its functional role in the process of EMT.

The combinatorial treatment of hesperetin and GSK-J4 at lower doses effectively prevented TGFβ-induced migration and the invasion of the PC-3 and LNCaP cells. There was synergism at concentrations where 50% of the cells were affected, whereas at higher concentrations, the drug combination was slightly antagonistic in PC-3 cells. In the LNCaP cells, synergism was seen in all combinations. The drugs could inhibit TGFβ-induced EMT by inhibiting both the canonical and non-canonical TGFβ signaling pathways. We need to look at other players in the non-canonical signaling pathway cascade to ensure the complete inhibition of TGFβ signaling, thereby inhibiting the migration and invasive capability of the prostate cancer cells effectively.

The effect of drug combinations on the H3K9me3 levels is contrasting in both cell lines, suggesting a difference in the genetic machinery of both cell lines, eventually leading to similar outcomes. The genes associated with the H3K9me3 modification after the combinatorial drug treatment need to be investigated to get a clearer understanding of the mechanism involved in both cell lines, which leads to the inhibition of TGFβ-induced EMT.

In conclusion, our study shows that GSK-J4 and hesperetin can be considered as effective therapeutic agents to treat prostate cancer (Figure 5). GSK-J4 inhibits cell viability in a dose-dependent manner. At lower concentrations, it can inhibit TGFβ-induced EMT by inhibiting TGFβ signaling. It also inhibits TGFβ-induced migration and invasion of prostate cancer cells. The effects are more pronounced in cancer lines with low tumorigenicity (LNCaP) compared to a highly tumorigenic cell line (PC-3). There are also a series of histone modifications associated with these observations, which need to be studied in further detail to better understand the underlying mechanisms involved. Our study also provides a new insight to combine an epigenetic inhibitor (GSKJ4) and a phytochemical (hesperetin) to prevent the TGFβ-induced EMT, as well as migration and invasion of prostate cancer cells. The epigenetic modifications are reversible, and phytochemicals are well known to be less toxic. This combination, therefore, will possibly be associated with lesser side effects, as compared to conventional chemotherapeutic drugs currently available for the treatment of prostate cancer.

## 4. Materials and Methods

### 4.1. Cells and Culture Conditions

PC3 and LNCaP cell lines were acquired from NCCS, Pune (Maharashtra, India). PC-3 cell line was procured in June 2021, and LNCaP cell line was procured in July 2021. The cell lines were authenticated using STR profiling and tested for mycoplasma using the Hoechst staining/PCR method at NCCS. There was a 100% match between the cell line samples and the ATCC STR profile database. The cell lines tested negative for mycoplasma contamination. The cells were grown in RPMI-1640 (HiMedia, Mumbai, Maharashtra, India) supplemented with 10% heat-inactivated fetal bovine serum (Gibco, Waltham, Massachusetts, USA) and 1% Antibiotic–Antimycotic solution (HiMedia, Mumbai, Maharashtra, India) at 37 °C in a 5% CO_2_ atmosphere. The cells were trypsinized at 80–90% confluency using 0.25% Trypsin-EDTA (HiMedia, Mumbai, Maharashtra, India).

### 4.2. MTT Assay

Cell proliferation, cytotoxicity, and cell viability were checked using the metathiazolyl tetrazolium (MTT) assay. To determine the effect of GSK-J4 (Sigma, St. Louis, MO, USA), the PC3 and LNCaP cells were exposed to different concentrations (1 µM, 2 µM, 4 µM, 6 µM, 8 µM, 10 µM, 25 µM, 50 µM, 100 µM) of GSK-J4 for 24 h. DMSO was used as solvent control. TGFβ treatment was given at a concentration of 5 ng/Ml. The assay was also used to check the effect of the combination of hesperetin and GSK-J4 on prostate cancer cells. MTT assay was carried out, as per standard protocol. The experiment was repeated thrice. The student’s *t*-test was carried out to calculate significance using GraphPad Prism 9.5.0 software.

### 4.3. Wound Healing Assay

Wound healing assay was carried out, as mentioned in the previous publication. For the combinatorial study, the concentration of hesperetin used was 100 µM, and that of GSK-J4 was 10 µM in PC-3 cells and 40 µM hesperetin and 4 µM GSK-J4 in LNCaP cells, both individually and in combination. The experiment was repeated thrice. The wound area was calculated using ImageJ 1.52a software. The student’s *t*-test was carried out to calculate significance using GraphPad Prism 9.5.0 software.

### 4.4. Western Blot Analysis

Cells were grown on 6-well plates at a seeding density of 2.5 × 10⁵ cells per well. These cells were starved in serum-free RPMI-1640 for 24 h. They were then treated with 5 ng/mL TGFβ and 20 µM GSK-J4, both individually and in combination. After 48 h of treatment, the cells were scraped in RIPA lysis buffer for protein extraction. Protein concentration was estimated using the Bradford assay. SDS-PAGE and Western blot were carried out as per standard protocol. Primary antibodies used were p-SMAD3 (Santa Cruz, Dallas, TX, USA), N-cadherin (Novus, St. Louis, MO, USA), vimentin (Novus, St. Louis, MO, USA), H3 (Invitrogen, Waltham, MA, USA), H3K9me3 (Santa Cruz, Dallas, TX, USA), H3K4me3 (Invitrogen, Waltham, MA, USA), H2K27me3 (Abcam, Cambridge, UK), and actin (Invitrogen, Waltham, MA, USA), and the secondary antibodies were Chicken Anti-Mouse IgG H&L (HRP) (Abcam, Cambridge, UK) and Chicken Anti-Rabbit IgG H&L (HRP) (Abcam, Cambridge, UK). The blot was developed using an ECL reagent (Thermo Scientific SuperSignal West Femto Maximum Sensitivity Substrate, Waltham, MA, USA). The experiment was repeated thrice. The statistical analysis was carried out using ImageJ 1.52a software. The student’s *t*-test was carried out to calculate significance using GraphPad Prism 9.5.0 software.

### 4.5. Invasion Assay

Cells were cultured to 75% confluency and starved in serum-free media for 24 h. After starvation, cells were trypsinized and washed twice in serum-free media. Approximately 30,000 cells were seeded with 500 µL serum-free media in the upper well of invasion transwell chambers coated with a Matrigel (BD BioCoat 8.0micron, Franklin Lakes, NJ, USA). Invasion assay was carried out, as per standard protocol. A minimum of five microscopic fields were counted from each membrane. The concentration of hesperetin used was 100 µM, and that of GSK-J4 was 10 µM, both individually and in combination. The experiment was repeated thrice. The cells were counted under the light microscope at 100× magnification for each sample using the ImageJ 1.52a software. Five microscopic fields were counted from each membrane. The student’s *t*-test was carried out to calculate significance using GraphPad Prism 9.5.0 software.

### 4.6. The qRT-PCR

PC3 cells were cultured in a T-75 flask until 70–80% confluency. The cells were then serum-starved for 24 h in a serum-free medium. After 24 h of incubation with 20 µM GSK-J4 alone, 5 ng/mL TGFβ alone, and the combination of TGFβ and GSK-J4, the total RNA was extracted using RNeasy Mini Kit (Qiagen, Hilden, Germany), according to the manufacturer’s protocol. DNA was removed from the samples by the RNAse-free DNAse set (Qiagen, Hilden, Germany). The mRNA was then reverse transcribed to cDNA (Eppendorf Mastercycler, Hamburg, Germany) with the help of an iScript cDNA Synthesis kit (Biorad, Hercules, CA, USA). Primers specific for E-cadherin, MLL1, SUV39H2, EZH2, and GAPDH were used (Table 1).

### 4.7. Experimental Design and Computerized Simulation Using Compusyn

To find out if two drugs are synergistic, additive, or antagonistic, it is important to find out the potency and shape of the dose–effect curve, which will give the IC50 (Dm) and slope (m) of each drug individually (m1, (Dm)1, m2 and (Dm)2), as well as in combination (m1,2, (Dm)1,2). These values can then be used to calculate the combination index (CI) value [31]. For the determination of these values, the CompuSyn software (free download from www.combosyn.com, accessed on 5 December 2022) was used. The software automatically plots the dose–effect curve and using the median–effect equation derived from the mass action law principle, the CI value is calculated [32]. The CI value is considered as a standard for determining if a drug combination is CI < 1, additive effect is CI = 1, and antagonism is CI > 1. Five different doses were chosen for both the drugs from MTT assays performed previously such that it included the IC50 dose of each drug. For hesperetin, five data points of 50, 100, 200, 300, and 400 µM were used. GSK-J4 treatment also included five data points of 5, 10, 20, 30, and 40 µM were taken. For combination, the diagonal constant ratio combination design proposed by Chou and Talalay was used [33]. Hesperetin and GSK-J4 were combined in a ratio of 10:1 as 50 + 5, 100 + 10, 200 + 20, 300 + 30, and 400 + 40, where both drug concentrations are represented in micromolars (µM). The experiment was repeated thrice. The entire CompuSyn report is available in the Appendix A.

## Figures and Tables

**Figure 1 ijms-24-01860-f001:**
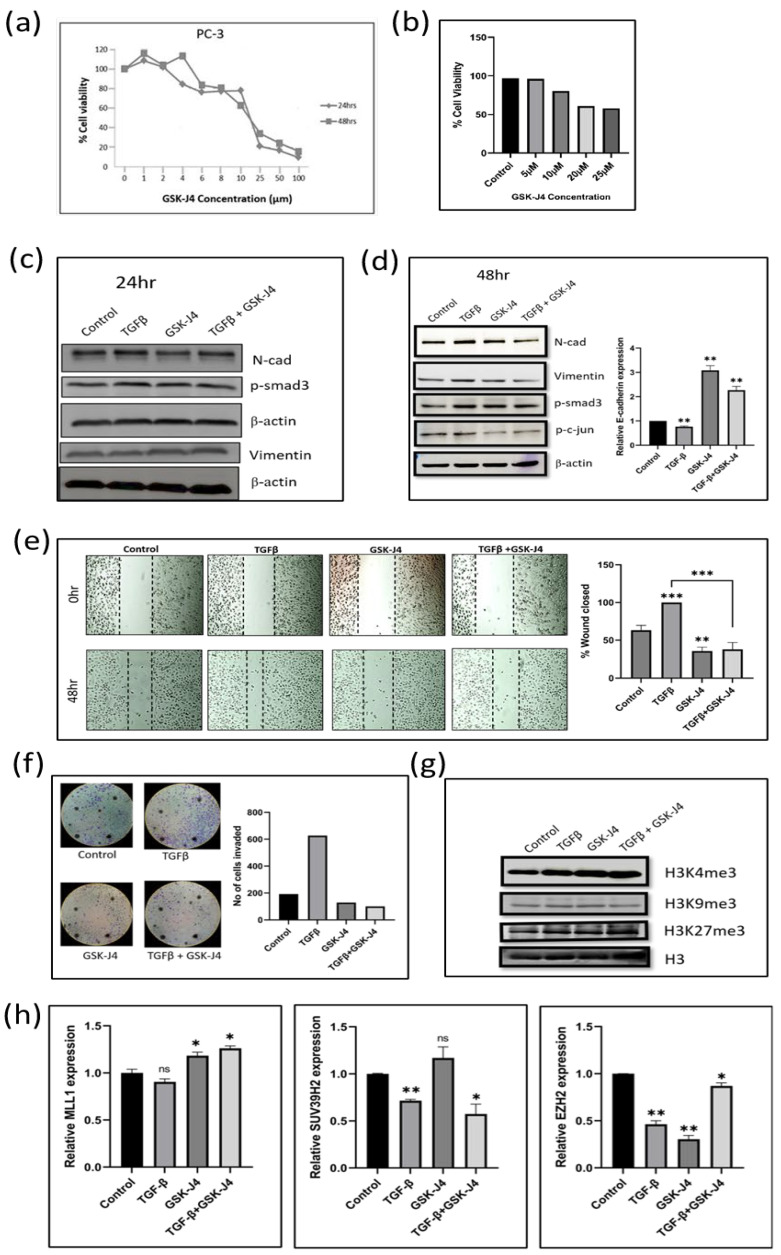
(**a**) Graph shows results of MTT assay in the PC3 cell line to determine the effect of GSK-J4 on cell proliferation. DMSO was taken as the vehicle control. (**b**) The bar graph represents the effect of GSK-J4 on cell viability based on Trypan blue dye uptake in PC-3 cells. DMSO was taken as the vehicle control. (**c**) Western blot image of N-cadherin, p-smad, and vimentin in PC-3 cell line after 24 h of treatment. β-Actin was taken as the loading control. (**d**) Western blot image of N-cadherin, p-smad3, p-c-jun, and vimentin in PC-3 cell line after 48 h of treatment. β-Actin was taken as the loading control. The graph represents the relative quantification of E-cad mRNA in PC3 cells upon treatment with GSK-J4 in the absence and presence of TGFβ. GAPDH was taken as the endogenous gene control, and the gene expression was normalized with respect to the control. n = 3 (**e**) Microscopic images of PC3 cells, as observed under a light microscope at a total magnification of 100×. For wound healing assay, the cells were treated with GSK-J4 in the presence and absence of TGFβ. The graph represents the % of wound area closed with respect to the 0 h time-point of each sample. The wound area was calculated using ImageJ 1.52a software (n = 3). (**f**) Invasion assay images of PC-3 cells taken under a light microscope at 100x magnification. The bar graph represents the number of cells that were invaded across the Matrigel. The cells were counted under the light microscope at a total magnification of 100× for each sample (n = 3) using the ImageJ 1.52a software. Five microscopic fields were counted from each membrane. (**g**) Western blot of H3K4me3, H3K9me3, and H3K27me3, after treatment with GSK-J4 in the presence and absence of TGFβ in PC-3 cells. Total histone (H3) was taken as the loading control. (**h**) The graphs represent the relative quantification of MLL1, SUV39H2, and EZH2 mRNA in PC3 cells upon treatment with GSK-J4 in the absence and presence of TGFβ. GAPDH was taken as the endogenous gene control, and the gene expression was normalized with respect to the control; n = 3, * *p* < 0.05, ** *p* < 0.01, *** *p* < 0.001.

**Figure 2 ijms-24-01860-f002:**
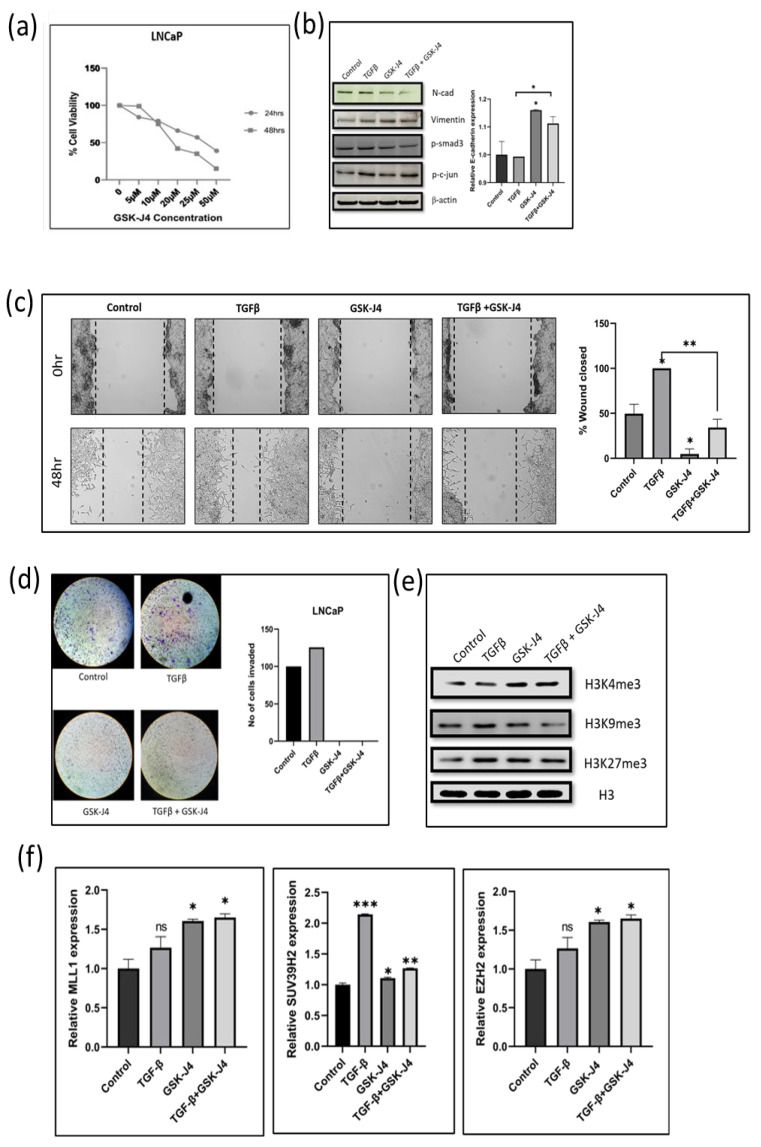
(**a**) Graph shows results of MTT assay in LNCaP cell line to determine the effect of GSK-J4 on cell proliferation. DMSO was taken as the vehicle control. (**b**) Western blot image of N-cadherin, p-smad, p-c-jun, and vimentin in LNCaP cell line after 48 h of treatment. β-Actin was taken as the loading control. The graph represents the relative quantification of E-cad mRNA in LNCaP cells upon treatment with GSK-J4 in the absence and presence of TGFβ. GAPDH was taken as the endogenous gene control, and the gene expression was normalized with respect to the control (n = 3). (**c**) Microscopic images of LNCaP cells, as observed under a light microscope at a total magnification of 100×. For wound healing assay, the cells were treated with GSK-J4 in the presence and absence of TGFβ. The graph represents the % of wound area closed with respect to the 0 h time-point of each sample. The wound area was calculated using ImageJ 1.52a software (n = 3). (**d**) Invasion assay images of LNCaP cells taken under a light microscope at 100× magnification. The bar graph represents the number of cells that were invaded across the Matrigel. The cells were counted under the light microscope at a total magnification of 100× for each sample using the ImageJ 1.52a software. Five microscopic fields were counted from each membrane. (**e**) Western blot of H3K4me3, H3K9me3, and H3K27me3 after treatment with GSK-J4 in the presence and absence of TGFβ in LNCaP cells. Total histone (H3) was taken as the loading control. (**f**) The graphs represent the relative quantification of MLL1, SUV39H2, and EZH2 mRNA in LNCaP cells upon treatment with GSK-J4 in the absence and presence of TGFβ. GAPDH was taken as the endogenous gene control, and the gene expression was normalized with respect to the control; n = 3, * *p* < 0.05, ** *p* < 0.01, *** *p* < 0.001.

**Figure 3 ijms-24-01860-f003:**
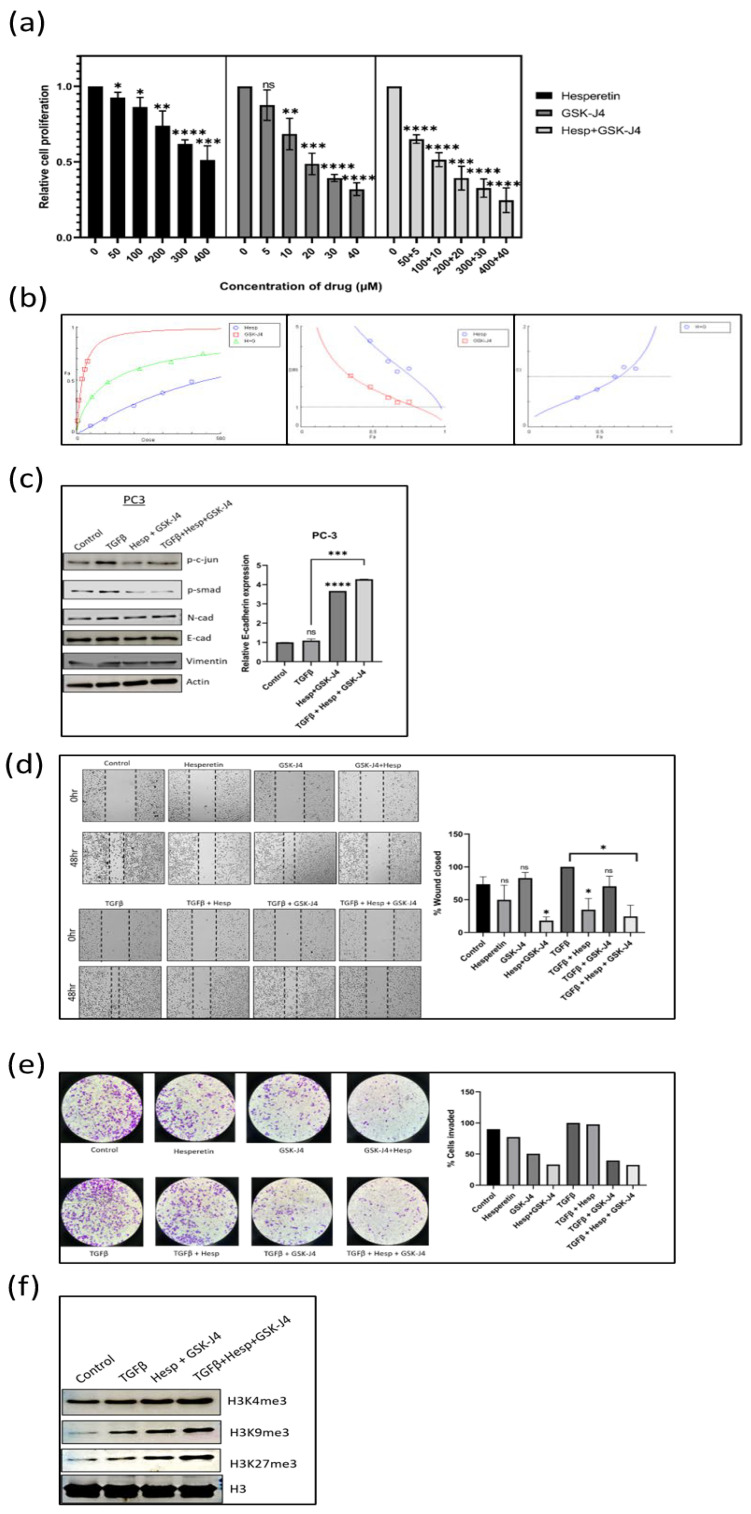
(**a**) MTT assay in the PC3 cell line to determine the effect of GSK-J4 and hesperetin alone and, in combination, on cell proliferation. DMSO was taken as the vehicle control. (**b**) Pharmaco-dynamic computerization and transformation after Compusyn analysis of hesperetin and GSK-J4 alone, as well as in combination, in PC-3 cells. The graphs indicate the dose–effect curve, CI plot, and DRI plot, respectively, for the combination of hesperetin and GSK-J4. (**c**) Western blot image of N-cadherin, p-smad, p-c-jun, and vimentin in PC-3 cell line after 48 h of treatment. β-Actin was taken as the loading control. The graph represents the relative quantification of E-cad mRNA in PC3 cells upon treatment with GSK-J4 in the absence and presence of TGFβ. GAPDH was taken as the endogenous gene control, and the gene expression was normalized with respect to the control (n = 3). (**d**) Microscopic images of PC3 cells, as observed under phase contrast microscope at a magnification of 100×. For wound healing assay, the cells were treated with GSK-J4 (10 µM) and hesperetin (100 µM) in the presence and absence of TGFβ both alone and in combination. The graph represents the % of wound area closed with respect to 0 hr time-point of each sample. The wound area was calculated using ImageJ 1.52a software (n = 3). (**e**) Invasion assay images of PC-3 cells taken under a light microscope at a total magnification of 100×. The cells were treated with GSK-J4 (10 µM) and hesperetin (100 µM) in the presence and absence of TGFβ both alone and in combination. The bar graph represents the number of cells that were invaded across the Matrigel. The cells were counted under the light microscope at 100× magnification for each sample using the ImageJ 1.52a software. Five microscopic fields were counted from each membrane. (**f**) Western blot of H3K4me3, H3K9me3, and H3K27me3 after treatment with GSK-J4 in the presence and absence of TGFβ in PC-3 cells. Total histone (H3) was taken as the loading control. * *p* < 0.05, ** *p* < 0.01, *** *p* < 0.001, **** *p* < 0.0001.

**Figure 4 ijms-24-01860-f004:**
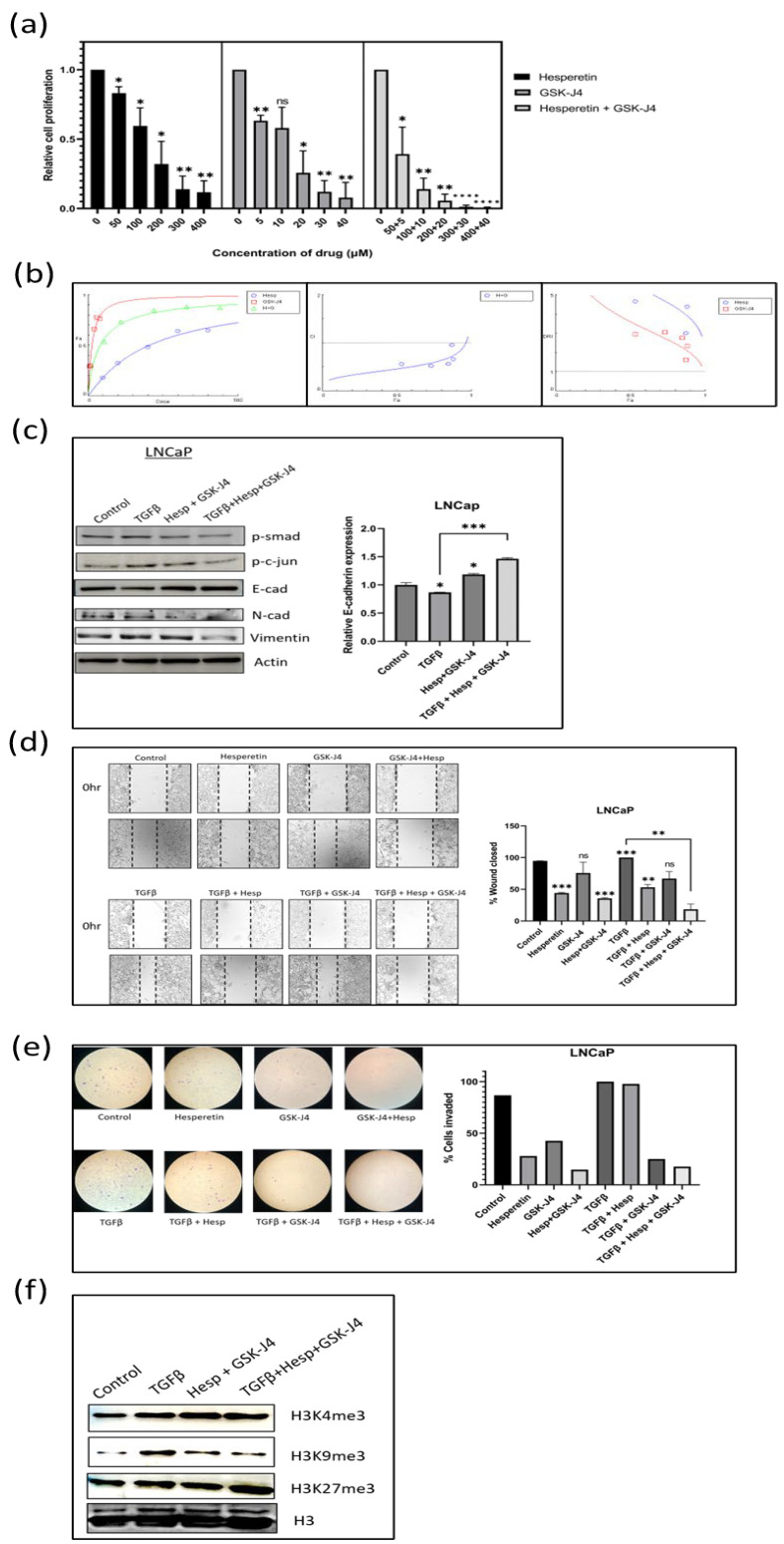
(**a**) MTT assay in LNCaP cell line to determine the effect of GSK-J4 and hesperetin alone and, in combination, on cell proliferation. DMSO was taken as the vehicle control. (**b**) Pharmaco-dynamic computerization and transformation after Compusyn analysis of hesperetin and GSK-J4 alone and in combination in LNCaP cells. The graphs indicate the dose–effect curve, CI plot, and DRI plot, respectively, for the combination of hesperetin and GSK-J4. (**c**) Western blot image of N-cadherin, p-smad, p-c-jun, and vimentin in LNCaP cell line after 48 h of treatment. β-actin was taken as the loading control. The graph represents the relative quantification of E-cad mRNA in LNCaP cells upon treatment with GSK-J4 in the absence and presence of TGFβ. GAPDH was taken as the endogenous gene control, and the gene expression was normalized with respect to the control (n = 3). (**d**) Microscopic images of LNCaP cells, as observed under a phase contrast microscope at a magnification of 100×. For wound healing assay, the cells were treated with GSK-J4 (10 µM) and hesperetin (100 µM) in the presence and absence of TGFβ both alone and in combination. The graph represents the % of wound area closed with respect to the 0 h time-point of each sample. The wound area was calculated using ImageJ 1.52a software (n = 3). (**e**) Invasion assay images of LNCaP cells taken under a light microscope at a total magnification of 100×. The cells were treated with GSK-J4 (10 µM) and hesperetin (100 µM) in the presence and absence of TGFβ both alone and in combination. The bar graph represents the number of cells that were invaded across the Matrigel. The cells were counted under the light microscope at 100× magnification for each sample using the ImageJ 1.52a software. Five microscopic fields were counted from each membrane. (**f**) Western blot of H3K4me3, H3K9me3, and H3K27me3 after treatment with GSK-J4 in the presence and absence of TGFβ in LNCaP cells. Total histone (H3) was taken as the loading control. * *p* < 0.05, ** *p* < 0.01, *** *p* < 0.001, **** *p* < 0.0001.

**Figure 5 ijms-24-01860-f005:**
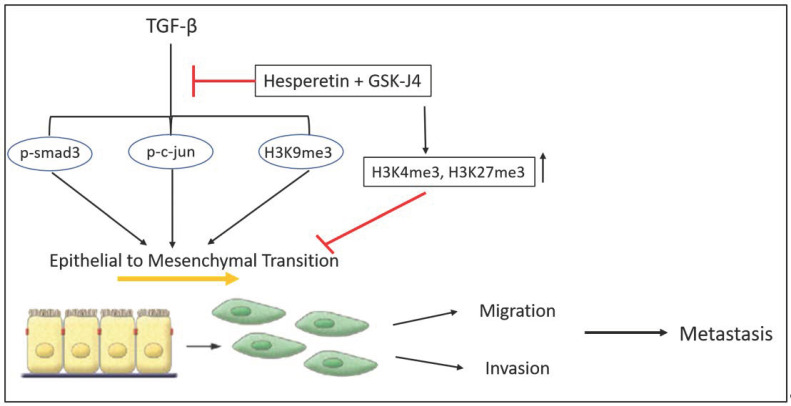
A schematic summary of the effects of combinatorial treatment of hesperetin and GSK-J4 on TGFβ-induced EMT and histone methylations in prostate cancer cells.

**Table 1 ijms-24-01860-t001:** The table represents the genes and their respective forward and reverse primer sequences used for qRT-PCR analysis.

Genes	Forward Primer	Reverse Primer
E-cadherin	5′-CGGGAATGCAGTTGAGGATC-3′	5′-AGGATGGTGTAAGCGATGGC-3′
N-cadherin	5′-ATTGGACCATCACTCGGCTTA-3′	5′-CACACTGGCAAACCTTCACG-3′
MLL1	5′-GAAGTGGTTCCTGAGAATGG-3′	5′-CACAGTCGGAGAGATCATTTAG-3′
SUV39H2	5′-ATTGATAACCTCGATACTCGTCTT-3′	5′-TCTCCAGAACCTTTCATTTGATAA-3′
EZH1	5′-CAATTCAAGCTGGCGAAGAG-3′	5′-CAAGACAGTGCCGCTACCA-3′
EZH2	5′-GCCAAGAGAGCCATCCAGAC-3′	5′-CCGACATACTTCAGGGCATCA-3′
GAPDH	5′-ATGTTCGTCATGGGTGTGAA-3′	5′-TGTGGTCATGAGTCCTTCCA-3′

The cDNA for the different samples was amplified (applied biosystems real-time PCR) using iTaq Universal SYBR Green Supermix (Biorad). The following were the cycling conditions: 30 s at 95 °C, followed by 40 Cycles of 15 s at 95 °C, 30 s at 56 °C, and 1 min at 72 °C. The experiment was repeated thrice. Data were collected and analyzed using QuantStudio Design and Analysis Software v1.5.1.

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
