# Peer review of "Combination Treatment of a Phytochemical and a Histone Demethylase Inhibitor—A Novel Approach towards Targeting TGFβ-Induced EMT, Invasion, and Migration in Prostate Cancer"

_ijms, 2023, doi:10.3390/ijms24031860_

Round 1

Reviewer 1 Report

1. If the authors refer to stem cell-like characteristics, they should specify the EMT type.

2. The authors should specify the transwell chambers used (at present it is impossible to distinguish if the assay performed was the invasion or chemotaxis assessment in individual subsections). If the chambers were not coated with Matrigel it was only chemotactic assay.

3.The literature in the Introduction section should be updated

4. p-c-Jun downregulation after combined treatment with TGFB,  Hesperetin and GSK-J4 (figure 3c) is not convincing

5. The EMT occurence in the fig 3c and 4c is also debatable

6. The western blot graph statistic is missing

7.In the discussion, the statement that the combination of drugs was able to inhibit TGFβ-induced EMT by inhibiting both the canonical and non-canonical TGFβ signaling pathways only taking into account p-Smad3 and p-c-Jun, which ensures the complete inhibition of TGFβ signaling seems to be a little bit exaggerated, particularly in the light of the previous comments

Author Response

Manuscript ID:  ijms-2112611

Title: Combination treatment of a phytochemical and a histone demethylase inhibitor - A novel approach towards targeting TGFβ induced EMT, Invasion, and Migration in Prostate Cancer.

Reviewer 1:

Comment 1: If the authors refer to stem cell-like characteristics, they should specify the EMT type.

Response: We have removed the statement regarding stem cell-like characteristics and also specified the EMT type in the introduction, as suggested by the reviewer on page no. 2, lines no. 54,55, and 56.

Comment 2: The authors should specify the transwell chambers used (at present, it is impossible to distinguish if the assay performed was the invasion or chemotaxis assessment in individual subsections). If the chambers were not coated with Matrigel, it was only chemotactic assay.

Response: We apologize for missing out on the chamber specifications. The chambers were indeed coated with a layer of matrigel, and hence the assay was performed for the assessment of the invasive capability of the cells. We have updated the information in the materials and methods section as per the suggestion on page no. 14 under the heading “Invasion Assay.”

Comment 3: The literature in the Introduction section should be updated.

Response: The literature in the introduction has been updated as suggested by the reviewer.

Comment 4: p-c-Jun downregulation after combined treatment with TGFB,  Hesperetin, and GSK-J4 (figure 3c) is not convincing.

Response: The downregulation of p-c-jun is significant when the treatment is given as compared to TGF? induction. The results are more clear in the quantification data of the respective blot image attached in Supplementary File 2 - Fig 3(c).

Comment 5: The EMT occurrence in the fig 3c and 4c is also debatable.

Response: There is a decrease in the mesenchymal markers though not drastic. The decrease is also clearly visible in the quantification images attached in supplementary file 2 - Fig 3c and Fig 4c. But the functional assays clearly indicate a significant inhibition of migration and invasion, which are an outcome of EMT inhibition.

Comment 6: The western blot graph statistic is missing.

Response: We have included the western blot graphs as a supplementary file 2, as suggested by the reviewer.

Comment 7: In the discussion, the statement that the combination of drugs was able to inhibit TGFβ-induced EMT by inhibiting both the canonical and non-canonical TGFβ signaling pathways only taking into account p-Smad3 and p-c-Jun, which ensures the complete inhibition of TGFβ signaling seems to be a little bit exaggerated, particularly in the light of the previous comments.

Response: We agree with the reviewer and have made changes to the statement accordingly. We have stated that “We need to look at other players in the non-canonical signaling pathway cascade to ensure the complete inhibition of TGFβ signaling, thereby inhibiting the migration and invasive capability of the prostate cancer cells effectively.” on page no. 12, Line 404.

Reviewer 2 Report

The article entitled "Combination treatment of a phytochemical and a histone demethylase inhibitor - A novel approach towards targeting TGFβ induced EMT, Invasion, and Migration in Prostate Cancer" provides new interesting insights into the value of therapeutic targeting of EMT (epithelial -mesenchymal- transition).in prostate cancer. The findings are significant but there are specific issues with the manuscript and work presentated that need to be addressed by the authors as follows:

1) The manuscript is too long and extensive technical details in the desrciption of the experiments (standard methodology) and the interpretation of the data need to be eliminated for clarity of flow.

2) A schematic diagram is recommended  (considering the complexity of the pathways targeted by the therapeutic combination) to indicate the signaling interactions between the diferent players and facilitate the appreciation of the results by the reader.

3) The study is largely desrciptive. The results presented are desrcibing the in vitro effects of the combination oof the inhibitors on the EMT phenotypic profile. There is no rigorous functional analysis as to the TGF-beta signaling effectors/their activation targeted by the drugs (such as Smade phosphorylation). A higher level mechanistic dissection is required.

4)  Another major weakness of the study is the lack of the in vivo component. The therapeutic antitumor effect of the drug combination in in vivo models ( animal models) in order to enable the extrapolation to a clinical setting of prostate cancer.

Author Response

Reviewer 2:

Comments to the Author

The article entitled "Combination treatment of a phytochemical and a histone demethylase inhibitor - A novel approach towards targeting TGFβ induced EMT, Invasion, and Migration in Prostate Cancer" provides new interesting insights into the value of therapeutic targeting of EMT (epithelial-mesenchymal- transition) in prostate cancer. The findings are significant, but there are specific issues with the manuscript and work presented that need to be addressed by the authors as follows:

Response: We thank you for the positive consideration of our work and for your comments. We will be happy to incorporate your valuable suggestions into our work.

Comment 1: The manuscript is too long, and extensive technical details in the description of the experiments (standard methodology) and the interpretation of the data need to be eliminated for clarity of flow.

Response: We have reduced the content in the material and methods section, as suggested by the reviewer.

Comment 2: A schematic diagram is recommended  (considering the complexity of the pathways targeted by the therapeutic combination) to indicate the signaling interactions between the different players and facilitate the appreciation of the results by the reader.

Response: We have included a schematic diagram below the discussion on Page no. 13, as suggested by the reviewer. The schematic is also attached below:

Comment 3: The study is largely descriptive. The results presented are describing the in vitro effects of the combination of the inhibitors on the EMT phenotypic profile. There is no rigorous functional analysis as to the TGF-beta signaling effectors/their activation targeted by the drugs (such as Smade phosphorylation). A higher level mechanistic dissection is required.

Response: The proteins such as Smad3 and c-jun are phosphorylated by TGF? induction. Moreover, Smad3 is one of the first downstream molecules to be phosphorylated by TGF? signaling. We have shown in our western blot data the upregulation of phosphorylation of Smad3 and c-jun in the presence of TGF? as well as after drug treatment. Other proteins like N-cadherin and Vimentin are the effector proteins that get upregulated due to TGF? signaling. Those have also been shown to be upregulated after TGF? induction.

Comment 4:  Another major weakness of the study is the lack of the in vivo component. The therapeutic antitumor effect of the drug combination in in-vivo models ( animal models) in order to enable the extrapolation to a clinical setting of prostate cancer.

Response: We do agree with the reviewer that the study needs to be carried out in in-vivo models to extrapolate the findings to a clinical setting of prostate cancer. This study has been carried out in-vitro as it is important first to check the effect of the drugs on human-derived cancer cells to make sure it is effective and feasible to take it further to an in-vivo setting which is labor- and cost-intensive. We are planning on taking it further on animal models to make sure that the combination treatment is both effective and non-toxic in-vivo.

Round 2

Reviewer 1 Report

1. Line 62: it would be better to write involve among others

2. In the supplementary file statistics are still missing

3.In the Materials and methods section there is no description of the statistical tools used.

4. The number of experimental repeats should be also clarified  each time

Author Response

Manuscript ID:  ijms-2112611

Title: Combination treatment of a phytochemical and a histone demethylase inhibitor - A novel approach towards targeting TGFβ induced EMT, Invasion, and Migration in Prostate Cancer.

Reviewer 1:

Comment 1. Line 62: it would be better to write involve among others

Response: We have incorporated the change as per the reviewer’s suggestion.

Comment 2. In the supplementary file statistics are still missing

Response: We apologize for missing out on the statistics in the graphs. We have added the graph statistics as per the reviewer’s suggestion.

Comment 3. In the Materials and methods section there is no description of the statistical tools used.

Response: The statistical tools have been mentioned in the figure legends. We have now mentioned them in the material and methods section as well.

Comment 4. The number of experimental repeats should be also clarified  each time

Response: We have mentioned the number of experimental repeats as well in the material and methods section, as suggested by the reviewer.

Reviewer 2 Report

The response by the authors has been insightful and appropriate. The work presented is novel and merits publication.

Author Response

Thanks for your appreciation.